# In Vitro Vascular-Protective Effects of a Tilapia By-Product Oligopeptide on Angiotensin II-Induced Hypertensive Endothelial Injury in HUVEC by Nrf2/NF-κB Pathways

**DOI:** 10.3390/md17070431

**Published:** 2019-07-23

**Authors:** Jiali Chen, Fang Gong, Mei-Fang Chen, Chengyong Li, Pengzhi Hong, Shengli Sun, Chunxia Zhou, Zhong-Ji Qian

**Affiliations:** 1College of Food Science and Technology, Guangdong Ocean University, Zhanjiang 524088, China; 2School of Chemistry and Environmental Science, Guangdong Ocean University, Zhanjiang 524088, China; 3Shenzhen Institute of Guangdong Ocean University, Shenzhen 518108, China

**Keywords:** tilapia, HUVEC, angiotensin II, NF-κB, Nrf2, endothelial dysfunction

## Abstract

Angiotensin II (Ang II) is closely involved in endothelial injury during the development of hypertension. In this study, the protective effects of the tilapia by-product oligopeptide Leu-Ser-Gly-Tyr-Gly-Pro (LSGYGP) on oxidative stress and endothelial injury in Angiotensin II (Ang II)-stimulated human umbilical vein endothelial cells (HUVEC) were evaluated. LSGYGP dose-dependently suppressed the fluorescence intensities of nitric oxide (NO) and reactive oxygen species (ROS), inhibited the nuclear factor-kappa B (NF-κB) pathway, and reduced inducible nitric oxide synthase (iNOS), cyclooxygenase-2 (COX-2), and endothelin-1 (ET-1) expression, as shown by western blot. In addition, it attenuated the expression of gamma-glutamyltransferase (GGT) and heme oxygenase 1 (HO-1), as well as increasing superoxide dismutase (SOD) and glutathione (GSH) expression through the nuclear factor erythroid 2-related factor 2 (Nrf2) pathway. Other experiments revealed that LSGYGP increased the apoptotic inhibition ratio between cleaved-caspase-3/procaspase-3, reduced expressions of pro-apoptotic ratio between Bcl-2/Bax, inhibited phosphorylation of mitogen-activated protein kinases (MAPK), and increased phosphorylation of the serine/threonine kinase (Akt) pathway. Furthermore, LSGYGP significantly decreased Ang II-induced DNA damage in a comet assay, and molecular docking results showed that the steady interaction between LSGYGP with NF-κB may be attributed to hydrogen bonds. These results suggest that this oligopeptide is effective in protecting against Ang II-induced HUVEC injury through the reduction of oxidative stress and alleviating endothelial damage. Thus, it has the potential for the therapeutic treatment of hypertension-associated diseases.

## 1. Introduction

Endothelial dysfunction is regarded as a predictor of cardiovascular diseases (CVD) and long-term clinical outcomes, such as heart disease, arteriosclerosis, stroke, kidney disease, and hypertension. Hypertension is a CVD which results in high death rates; however, its pathogenesis and precise mechanism, at present, remain unknown. There is a common perception that hypertension is connected with vascular endothelial dysfunction caused by inflammation cytokines and oxidative stress in vascular endothelial cells [1,2]. Research has shown that inflammatory stimulation in endothelial cells could be induced by the extracellular stimulators angiotensin II (Ang II) [3], tumor necrosis factor-alpha (TNF-α) [4], and lipopolysaccharide (LPS) [5] which may cause further vascular endothelial dysfunction. Ang II is a crucial active peptide produced in the renin–angiotensin system (RAS), which is responsible for regulating downstream cellular factors and physiological responses and, also, directly induces vascular injury through activating inflammation and oxidative stress [6]. Moreover, it has a dominant position in the phase of vascular normal physiology and disease taking part in endothelial damage [7]. It has been reported that other vascular dysfunction-related factors, such as reactive oxygen species (ROS), vasoconstrictor endothelin-1 (ET-1), and nitric oxide (NO), are induced by Ang II [8,9,10].

ROS is a vital bio-molecular factor, relevantly connected with the damage of endothelial cells and endothelial dysfunction, which participate in the development of CVDs [11]. It promotes the expression levels of anti-oxidant enzymes and inflammatory cytokines, including gamma-glutamyltransferase (GGT), glutathione (GSH), nitric oxide synthase (iNOS), superoxide dismutase (SOD), and cyclooxygenase-2 (COX-2) [12,13], resulting in the activation of the nuclear factor-kappa B (NF-κB) pathway, further activating oxidative stress by the downstream nuclear erythroid 2-related factor 2 (Nrf2) pathway [14,15]. Among these, the Nrf2 pathway, as a key controller of the redox homeostasis gene regulatory network [16], has been shown to be the most critical pathway against oxidative stress [17]. Heme oxygenase 1 (HO-1) functions as a downstream effector of the Nrf2 pathway and as vital antioxidant enzymes to suppress oxidative stress [18]. Moreover, under induction of oxidative stress and nitric oxide, it is mainly regulated by the activation of Nrf-2 and MAPK/ERK pathways in vascular and endothelial cells [19]. Furthermore, oxidative stress and inflammation induced by Ang II causes endothelial damage and even cell death [7], which is relevant to classic apoptotic pathways, including the Bcl-2 family (anti-apoptotic protein Bcl-2 and pro-apoptotic protein Bax) and the serine/threonine kinase (Akt) pathway [20].

Recently, some studies have reported natural compounds which suppress Ang II-induced hypertensive injury [3,6,21,22]; however, little is known about marine fish resources. A review [23] summarized that marine resources broaden chemical space to be explored in the pharmaceutical market. Especially with bioactive peptide, they linked antibody drugs and small molecules due to the advantage that peptides possess the properties of antibodies and small molecules. Tilapia (*Oreochromis niloticus*) plays an important role in aquaculture, expansion of its production in Asia and Africa has increased, which is partially attributable to the high protein content. It’s reported that during the process of tilapia filleting, other parts of the fish are wasted, for example, the frame and skin5 may contain approximately 80% protein content, but are underutilized [24]. Gelatin peptides have been widely used in food, cosmetics, and biomedical industries, due to their low molecular weight, high absorption rate, and bioavailability, as well as their antioxidant and anti-hypertensive functional and biological properties [25]. The oligopeptide Leu-Ser-Gly-Tyr-Gly-Pro (LSGYGP) has been purified completely from tilapia skin gelatin hydrolysates, with molecular weight 592.26 Da [26]. According to former studies, this oligopeptide has good antioxidant and anti-photoaging activities [27], but, to our knowledge, the other activities of LSGYGP have not been reported. This study is aimed to investigate whether it is able to protect against cardiovascular injury.

Endothelial cells are directly correlated prominent cells in CVD and, so, it is a good strategy to prevent cardiovascular diseases by inhibiting the stimulation of activated endothelial cells in order to improve excessive inflammatory response in diseases. Therefore, this study focuses on the cytoprotective effects of LSGYGP on Ang II-induced human umbilical vein endothelial cells (HUVEC) injury and the underlying mechanisms, including cytokine expressions of the Nrf2, NF-κB, MAPK, and Akt pathways.

## 2. Results

### 2.1. Cytoprotective Effect of LSGYGP on Ang II-Stimulated HUVEC

The cytotoxic concentration (10, 20, 50, and 100 μM) of LSGYGP (Figure 1a) and the evaluation of the protective effects of LSGYGP in Ang II-stimulated HUVEC were determined using an MTT assay. As shown in Figure 1b, at the concentrations tested (10–100 μM), LSGYGP did not affect cell viability, and LSGYGP treatment showed a cytoprotective effect on Ang II-stimulated HUVEC in a dose-dependent manner (Figure 1c).

### 2.2. LSGYGP against NO and ROS Production

As Figure 2a shows, the Ang II group had an obvious effect on NO and ROS levels, as compared to the untreated control. Remarkably, 10 μM treatment with LSGYGP decreased the Ang II-induced increase in NO and ROS generation. LSGYGP significantly attenuates NO and ROS production in a dose-dependent manner.

### 2.3. Effect of LSGYGP on the Nrf2 Pathway

Nrf2 is a major signaling pathway in oxidative stress reactions. As shown in Figure 3, reduction of SOD, GSH, Nrf2, and HO-1 proteins were discovered in the Ang II-induced HUVEC treatment group; however, a significant restoration was shown when treated with LSGYGP. Compared to the untreated control, Ang II treatment resulted in a significant augmentation of GGT. However, treatment with 50 μM LSGYGP diminished the Ang II-induced increase in GGT expression level. These findings indicate that LSGYGP increased antioxidant enzymes and inhibited the Nrf2 pathway.

### 2.4. Effect of LSGYGP on MAPK Pathway

Western blot was used to detect phosphorylation of JNK, ERK, and p38. Moreover, Ang II markedly increased protein expression of phosphorylated p38, JNK, and ERK, as compared to the non-Ang II-treated control. However, the above changes were both markedly reversed by 100 μM treatment with LSGYGP (Figure 4b). The above results indicate that, while Ang II markedly activates endothelial oxidative stress, LSGYGP treatment may attenuate this oxidative damage through MAPK pathways.

### 2.5. Effect of LSGYGP on Inflammatory Factor and NF-κB Pathway

As LSGYGP could down-regulate NO production (as per the above summarized results), western blot was used to analyze whether LSGYGP decreased NO expression through iNOS down-regulation (Figure 5a). As shown in Figure 5b, the reduction level of iNOS was consistent with the result of NO production measured in LSGYGP treatment. Furthermore, COX-2 and ET-1 levels were also investigated. As depicted in Figure 5b, expression of COX-2 and ET-1 mainly increased after exposure to Ang II, but treatment of 10 μM LSGYGP markedly blocked the Ang II induced changes. Moreover, the results revealed that LSGYGP may inhibit expression and translocation of NF-κB p65 by suppressing the phosphorylation of IκBα.

Immunocytochemistry and electrophoretic mobility shift assay (EMSA) were applied to further investigate the translocation of NF-κB p65 in Ang II-stimulated HUVEC. Observing images in Figure 5c, the NF-κB p65 sub-unit was transported to the nucleus after Ang II stimulation, as indicated by the NF-κB p65 protein level in the western blot test. Employing image analysis, treatment with Ang II presented a significant increase in the DNA-binding activity of NF-κB, whereas treatment with LSGYGP significantly reduced the Ang II-induced DNA-binding activity of NF-κB (Figure 5d,e), this revealed that p65 might enter the nucleus based on DNA-binding activity.

With the above results, we can identify that Ang II treatment activated endothelial oxidative stress and may cause inflammatory injury, which caused the expression of the relative inflammatory factors and translocation of the NF-κB pathway.

### 2.6. Effect of LSGYGP on the Akt Pathway

Figure 6 shows a marked augmentation of the expression of Bax and cleaved-caspase-3 (c-caspase-3) and down-regulation of the proliferation-related p-Akt and anti-apoptosis protein Bcl-2 in the Ang II-stimulated group. These results were both reversed by LSGYGP treatment in a dose-dependent manner. The expression of procaspase-3 did not change in any treatment. The effects of LSGYGP on the Bcl-2 family and Akt pathway could explain its vascular-protective effects on endothelial injury.

### 2.7. DNA Damage in Comet Assay

In Figure 7a, no visible comets were observed in normal cells. Ang II treatment produced significantly long “comet tails”, while 10 μM LSGYGP treatment significantly reduced the comet tail induced by Ang II; the lengths of the comet tails decreased in a dose-dependent manner with LSGYGP treatment (Figure 7b). The above results suggested that Ang II treatment may cause inflammatory injury, which further results in the expression of the relative apoptosis factor and reduction of p-Akt; possibly even DNA damage. LSGYGP reversed these phenomena in a vascular-protective role.

### 2.8. Docking Results of LSGYGP with NF-κB

Molecular docking results revealed that LSGYGP connected with NF-κB in a steady interaction (Figure 8a). The CDOCKER results of the NF-κB–LSGYGP combination can evaluate the rationality of side-chain backbone interactions (−5.557 kcal/mol), which is shown at 2.0 Å resolution in Figure 8b (as can be seen, the first way is the best). LSGYGP generated 10 hydrogen bonds, with bond lengths of 5.42 Å, 6.41 Å, 4.83 Å, 5.35 Å, 4.74 Å, 4.61 Å, 3.57 Å, 5.24 Å, 5.81 Å, and 5.46 Å with Lys79, Lys79, Gln220, Gln29, Gln29, Met279, Glu282, Lys221, Lys221, and Lys221, respectively, in Table 1.

## 3. Discussion

Recently, the evidence from a great amount of research supports the idea that chronic vascular disease, such as atherosclerosis, hypertension, stroke, and so on, may be attributed to specific endothelial dysfunction for oxidative stress and inflammatory response [28,29]. Endothelial dysfunction of oxidative stress possesses a critical role in CVD, with an increasing amount of evidence [30]. Especially cumulating in the vascular endothelium, Ang II levels are frequently elevated at the initiation and in the progression of hypertension, upon which certain endotheliocytes undergo oxidative stress, which gives rise to endothelial dysfunction [31]. Some studies have provided evidence that endothelial oxidative stress promotes vascular cell apoptosis and increases inflammatory cytokine expression, causing the failure, relaxation, or dilation of arteries, leading to increased tension of the arterial wall [32], which may cause the progression of hypertension. Our study conducted a series of experiments to verify this theory and found that the tilapia by-product peptide LSGYGP can play a protective role and reverse these phenomena.

The ROS we discovered in Ang II-induced HUVEC is a common mediator of endothelial dysfunction and vascular inflammation in the cardiovascular system [33]. According to former studies, Ang II may result in ROS formation by activating nicotinamide adenine dinucleotide/triphosphopyridine nucleotide (NADH/NADPH) oxidases [31,34,35,36], and the result of ROS fluorescence of our study gave the same result (Figure 2). On one hand, the augmentation of ROS in oxidative stress, with the ability to stimulate the expression of pro-inflammatory factors [37], may further lead to endothelial dysfunction, lipid oxidation, and inflammatory responses [38]. The imbalance between NO with ET-1 is usually regarded as a predictor of hypertension. NO, as a vasoactive substance, might decrease endothelial cell activation through the mechanism of reducing NF-κB activation [39]. From the fluorescence results, it was clearly verified that LSGYGP reduced the Ang II-stimulated levels of NO production. Similarly, the Ang II-stimulated production of ROS was reduced by LSGYGP, as shown by the DCF fluorescence intensity in Figure 2. It has been reported that a great mass of inhibitors of NF-κB activation work by suppressing IκBα phosphorylation and degradation [40]. The results, in terms of the NF-κB pathway, detected phosphorylation of NF-κB in accordance with the above theory. Subsequently, LSGYGP treatment reduced this DNA-binding activity of NF-κB in a dose-dependent manner (Figure 5d).

On the other hand, oxidative stress could activate Nrf2, which is present in low levels under normal conditions, into the cell nucleus. HO-1 expression is regulated by Nrf2 levels, which leads to LSGYGP dose-dependently protecting from oxidative stress injury in vitro. Therefore, further study is needed on the effect of LSGYGP on the Ang II-mediated Nrf2 pathway and observing its relative protein expression. It has been shown that Nrf2, with stimuli, up-regulated the transcription of antioxidant enzymes [41]; the result of SOD in LSGYGP treatment in our study reversed it (Figure 3). Several studies have reported that Ang II activates the family JNK, ERK, and p38 of MAPK, which can result in the activation of NF-κB by phosphorylation and degradation of IκBα [42,43]. We found that LSGYGP had the ability to evidently decrease the pro-inflammatory factors iNOS and COX-2, suppress NF-κB phosphorylation and translocation, suppress MAPK phosphorylation, and suppress the relative protein expression in the Nrf2 pathway in Ang II-mediated HUVEC (Figure 3, Figure 4 and Figure 5), which presented the same protective effects as Tao’s osthole [6]. All in all, these results verify the theory that Ang II-induces ROS production and the NF-κB pathway, and thus activates the subsequent oxidative stress pathways, which both give rise to vascular endothelial injury, leading to endothelial dysfunction and, subsequently, CVD [44].

Yusuke et al. [45] suggested that the Ang II acts in a wide pathway-inhibitive role mechanism, which may be partly traced back to “intra-cellular cross talk” between Ang II and other inflammatory second mediators (NO, ET-1, iNOS, and COX-2) through transcription factor activation (e.g., NF-κB and Nrf2). In our study, LSGYGP suppressed those pathways by mediation of Ang II; we go deeper into the potential mechanisms of this result next. The results of this study suggest the restored expression of the proliferation-associated proteins p-Akt and Bcl-2 after LSGYGP treatment, as well as a marked down-regulation of caspase-3 and Bax (as observed in Figure 6), which is consistent with Shan [46]. The results of investigating the Akt pathway and the comet assay (Figure 7) suggest that oxidative stress further results in endothelial damage, and even cell death, to some degree.

The molecular docking results indicate that LSGYGP could interact with NF-κB, as a stable complex, by hydrogen bonds. A binding energy value of −5.557 kcal/mol in the optimal spatial structure (Figure 8a) was shown by the presence of the hydrogen bond, such that LSGYGP has a strong affinity toward NF-κB. Hydrophobic interactions have been regarded as the most important non-covalent force in the literature, and have been shown to be responsible for multiple phenomena, including the binding of enzymes to substrates, folding of proteins, and structure stabilization of proteins [47]. In the LSGYGP–NF-κB interaction, most of the amino acid interactions are attributed to the hydrophobic amino acids of LSGYGP. It is the amino acids Leu and Pro of LSGYGP that may contribute to the exhibition of inhibitory activity in Figure 8b, with the strong binding effect of hydrophobic interaction improving the binding affinity between NF-κB and LSGYGP. These docking studies imply that LSGYGP alleviates oxidative stress through inhibiting the activation of NF-κB.

This study has firstly reported that LSGYGP acts in a cytoprotective role against Ang II-induced oxidative stress and inflammation. LSGYGP altered the related protein expressions of the NF-κB/Nrf2 signaling pathways and reversed Ang II-induced cell endothelial injury for the first time. The entire signaling pathway of this study is shown in Figure 9. Considering all of the results, we may conclude that LSGYGP effectively attenuated Ang II-stimulated cellular injury by activation of the NF-κB/Nrf2/MAPK/Akt pathways, at both the cellular and molecular levels. This study has also illustrated that Ang II, through oxidative stress, inflammation, and apoptosis pathways, could contribute to injury events in endothelial dysfunction, especially in terms of hypertension and atherosclerosis, but future studies are still necessary to determine the specific receptors involved. Future research should further discuss and confirm that how Ang II to binds the angiotensin type 1-receptor (AT1R) and activates membrane-bound NAD(P)H oxidase for the formation of ROS; how oligopeptide LSGYGP to binds with AT1R through determinate the location for GABARAP. Besides, bioavailability and transport mechanisms of oligopeptide are remains to research for following bioactivity design and application of peptide drugs, these are the key concerns attempted to overcome by using different design strategies in the future.

## 4. Materials and Methods

### 4.1. Materials

Ang II, 3-(4,5-Dimethylthiazol-2-yl)-2,5-diphenyltetrazolium bromide (MTT), 2,7-dichlorodihydrofluorescein diacetate (DCFH-DA), Dimethyl sulfoxide (DMSO), and 4′,6-diamidino-2-phenylindole (DAPI) were provided by Sigma-Aldrich (St. Louis, MO, USA). LSGYGP was ordered from Hangzhou Dangang Biotechnology Co., Ltd. (Hangzhou, China) with 99.8% purity. The BCA protein assay kit was provided by Thermo Fisher Scientific, Inc. (Waltham, MA, USA). The following antibodies were purchased from Santa Cruz Biotechnology (Santa Cruz, CA, USA): Mouse polyclonal antibodies, including iNOS (sc-7271), COX-2 (sc-19999), SOD (sc-271014), GGT (sc-100746), GSH (sc-71155), Nrf2 (sc-365949), Keap1 (sc-365626), HO-1 (sc-136960), NF-κB p65 (sc-8008), NF-κB p-p65 (sc-136548), IκB-α (sc-1643), p-IκB-α (sc-8404), p-p38 (sc-166182), p-ERK (sc-81492), JNK (sc-7345), p-JNK (sc-6254), and glyceraldehyde-3-phosphate dehydrogenase (GAPDH) (sc-47724); rabbit polyclonal antibodies (ERK, sc-94; p38, sc-535); secondary antibodies, such as goat anti-mouse IgG-HRP (sc-2005) and goat anti-rabbit IgG-HRP (sc-2004). Green fluorescence secondary antibody (Dylight 488, A23220) was acquired from Abbkine (Redlands, CA, USA). 3-amino,4-aminomethyl-2′,7′-difluorescein diacetate (DAF-FM DA) and a chemiluminescent EMSA Kit were purchased from Beyotime Biotechnology (Shanghai, China). All other unmentioned reagents were of analysis grade.

### 4.2. Cell Culture

Human umbilical vein cells (HUVEC) were obtained from Bena Culture Collection Co., Ltd. (Beijing, China) and grown in Dulbecco’s modified Eagle’s medium (DMEM): 10% fetal bovine serum (FBS) and 1% penicillin-streptomycin in 5% CO_2_ at 37 °C.

### 4.3. Cell Viability Assay

Cytotoxicity was evaluated by MTT assay. The HUVEC were seeded in 96-well plates (1 × 10^4^ cells/well), pre-treated with LSGYGP (10, 20, 50, and 100 μM; 1 h) and, afterwards, by Ang II (1 μM; 24 h). After removing the cell culture, MTT solution (1 mg/mL; 100 μL) was added to each well for 4 h. Finally, the supernatants were removed and 100 µL DMSO was added to dissolve the formazan crystal. Absorbance at 540 nm was measured with a microplate reader (BioTek, Winooski, VT, USA).

### 4.4. Determination of NO and ROS

Productions of NO and intracellular ROS were assessed by measuring the fluorescence intensities of DAF-FM DA and DCFH-DA, respectively. Briefly, the cells were seeded in a 24-well plate (1 × 10^4^ cells/well), pre-treated with LSGYGP (10, 50, and 100 μM; 1 h), and then incubated with Ang II (1 μM; 24 h). The cells were washed in phosphate buffered saline (PBS) three times, subsequently being loaded with fluorochrome (5 μM DAF-FM DA, 20 min; 10 μM DCFH-DA, 30 min) at 37 °C in a CO_2_ incubator. After being rinsed again, productions of NO and ROS were observed with a fluorescence microscope (Olympus, Tokyo, Japan).

### 4.5. Western Blot

The HUVEC were cultured in 6-well plates (5 × 10^6^ cells/well). The cells were pre-treated with LSGYGP (10, 50, and 100 μM) for 1 h and subsequently treated with Ang II (1 μM) for 24 h. After treatment, the cells were washed with pre-cooled PBS, harvested by scraping, and lysed using lysis buffer on ice for 30 min. After centrifugation for 10 min at 12,000× *g*, the supernatants were collected to determine protein concentration using the Pierce BCA Protein Assay Kit. Equal amounts of protein (20–40 μg) were heated with pre-stained markers at 95 °C for 10 min, then were electrophoretically examined using a 10% sodium dodecyl sulfate-polyacrylamide gel electrophoresis (SDS-PAGE), then transferred onto nitrocellulose (NC) filter membranes (Amersham, USA). Non-specific binding sites were blocked with 5% skim milk in Tris-buffered saline Tween-20 (TBST) at room temperature for 3 h, and then incubated overnight at 4 °C with primary antibody (1:500). After being washed with TBST (4 times, 10 min/time), the membranes, following used secondary antibody (1:5000), were treated with horseradish peroxidase (HRP). Blotted antibody signals were detected with an enhanced chemiluminescence (ECL) system (Syngene, Cambridge, UK).

### 4.6. Immunocytochemistry

The HUVEC were seeded in 24-well plates (5 × 10^4^ cells/well) in advance, treated as described above, and were then harvested. After washing thrice with PBS buffer, the cells were fixed in phosphate buffer solution contained 4% paraformaldehyde (4 °C, 20 min). Then, permeabilization was carried out by using 0.2% Triton X-100 in PBS followed by incubation (4 °C, 10 min). Cells were then blocked with 5% bovine serum albumin (BSA) in PBS, removed and directly incubated overnight at 4 °C with anti-p65 antibody (1:100). After removing the primary antibody, the cells were washed again and incubated, in the dark and at room temperature for 3 h, with the corresponding Goat Anti-Rabbit IgG secondary antibody (1:500; Abbkine, CA, USA). Finally, the nuclei were stained using DAPI (100 ng/mL) for 5 min. The images were then observed under an inverted fluorescence microscope (Olympus, Tokyo, Japan).

### 4.7. EMSA Assay

The HUVEC were cultured in 6-well plates (1 × 10^6^ cells/well), treated as described above, and were then harvested. Nuclear extracts were collected, according to the manufacturer’s instructions of the Nuclear and Cytoplasmic Extraction Kit from Beyotime Biotechnology (Shanghai, China). Protein concentration was determined using the Pierce BCA Protein Assay Kit. The NF-kB probe was 5′-AGT TGA GGG GAC TTT CCC AGG C-3′ which reacted with 5 μg nuclear protein. The mixture was then electrophoretically separated on a 6.5% polyacrylamide gel in 0.5X Tris-borate buffer (100 V; 1 h) and transferred onto positive-charge nylon membranes with 0.5 × TBE running buffer for 50 min at 300 mA. The membrane was cross-linked for 15 min under UV light, blocked with blocking solution containing streptavidin-HRP conjugate, and finally visualized using Chemiluminescent EMSA Kit, according to the manufacturer’s instructions.

### 4.8. Comet Assay

The comet assay was performed as described previously [48] to evaluate DNA strand breakdown. Cells were treated as described above and then suspended in PBS (1 × 10^5^ cells/mL). Briefly, 1% low melting point agarose (LMA, 80 µL) was added to the cells (200 cells/µL, 20 µL), which were then put on a slide pre-coated with 0.8% normal agarose (NMA, 100 µL) dissolved in PBS, and covered immediately with a coverslip (4 °C, 15 min). After solidification, the coverslip was cautiously removed. The cells were lysed in a pre-chilled lysis solution containing 2.5 M NaCl, 100 mM Na_2_EDTA, 10 mM Tris, 200 mM NaOH, 1% sodium lauroyl sarcosinate, and 1% Triton X-100 at pH 10 (4 °C; 90 min). The slides were placed in an electrophoresis chamber with an alkaline electrophoresis solution (200 mM NaOH and 1 mM Na_2_EDTA, pH > 13) for unwinding and expression of alkali-labile sites (30 min) and were subsequently electrophoresed (25 V and 300 mA, 20 min) with aim to draw the negatively charged DNA toward the anode. After neutralization, the cells were stained with DAPI (50 µg/mL, 20 µL) in the dark for 5 min. They were then observed under an invert fluorescence microscope (Olympus, Tokyo, Japan) the DNA damage (average tail length) was quantified using the CASP software to analyze the comet images.

### 4.9. Molecular Docking

The structure of LSGYGP was drawn using the Chemdraw software (Chemdraw, PerkinElmer Informatics, Boston, MA, USA). The three-dimensional (3D) crystal structure of NF-κB (PDB: 1IKN) was downloaded from the Protein Data Bank (PDB) (http://www.rcsb.org/pdb/). The CDOCKER algorithm in the Discovery Studio (DS) 3.5 software (Accelrys Software Inc., San Diego, CA, USA) was used to simulate the protein–ligand interaction. The high-molecular dynamics method was utilized to randomly search for small molecule (LSGYGP) conformations, and simulated annealing was used to optimize each conformation of the active site of the receptor (NF-κB).

### 4.10. Statistical Analysis

All analyses were carried out on triplicate samples. The GraphPad Prism 5.0 software (GraphPad Prism Software Inc., La Jolla, CA, USA) was used for the statistical analysis. Multiple-group comparisons were evaluated by one-way ANOVA, accompanied by Dunnett’s multiple comparison test for group comparison.

## 5. Conclusions

In conclusion, Ang II is a well-known powerful inducer of oxidative stress and inflammatory responses in cardiovascular tissues, resulting in atherosclerosis and hypertension. LSGYGP inhibited Ang II-stimulated oxidative stress and vascular endothelial dysfunction; down-regulated iNOS and COX-2 by suppressing the NF-κB pathway; up-regulated SOD and GSH by suppressing the phosphorylation of MAPK; and up-regulated HO-1 by the Nrf2 pathway. Simultaneously, down-regulation of the production of NO and ROS by LSGYGP was involved in the protective process, which could help to protect vascular function. Docking results suggest that LSGYGP may steadily connect with NF-κB by hydrogen bond interactions. This study revealed that LSGYGP has protective effects against the inflammation, oxidant stress, and apoptosis induced by Ang II in HUVEC, which may provide one of the underlying mechanisms for the treatment of Ang II-stimulated endothelial dysfunction and, thus, has potential for application in curing hypertensive disorders.

## Figures and Tables

**Figure 1 marinedrugs-17-00431-f001:**
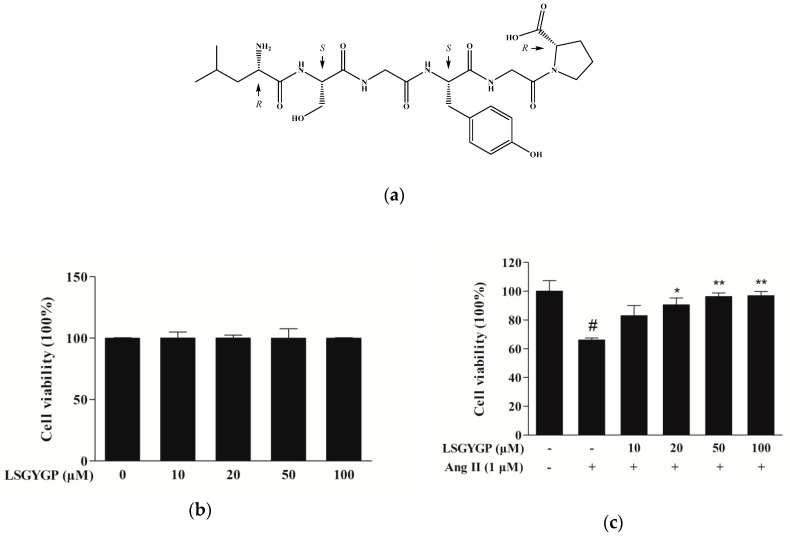
(**a**) Structure of Leu-Ser-Gly-Tyr-Gly-Pro (LSGYGP). Effect of LSGYGP on the viability of human umbilical vein endothelial cells (HUVEC). (**b**) and protective effect of LSGYGP on the viability of Ang II-treated HUVEC. (**c**) Cells were exposed to varying concentrations (10, 20, 50, and 100 μM) of LSGYGP and the cell viability was assessed by MTT assay. # *p* < 0.001, compared with blank group (untreated cells); * *p* < 0.05, ** *p* < 0.01, compared with control group (Ang II-treated cells).

**Figure 2 marinedrugs-17-00431-f002:**
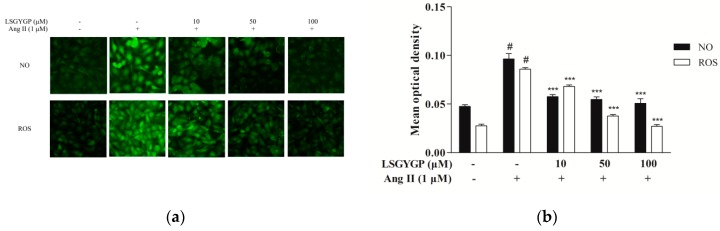
(**a**) Representative 3-amino,4-aminomethyl-2′,7′-difluorescein diacetate (DAF-FM DA) fluorescent images and 2,7-dichlorodihydrofluorescein diacetate (DCFH-DA) fluorescent images of LSGYGP in HUVEC. Mean optical density analysis of cellular DAF-FM DA staining (green fluorescence; an indicator of production of NO) and mean optical density analysis of cellular DCFH-DA staining (green fluorescence; an indicator of production of ROS). (**b**) Mean optical density analysis of fluorescent images. # *p* < 0.001, compared with blank group (untreated cells); *** *p* < 0.001, compared with control group (Ang II-treated cells).

**Figure 3 marinedrugs-17-00431-f003:**
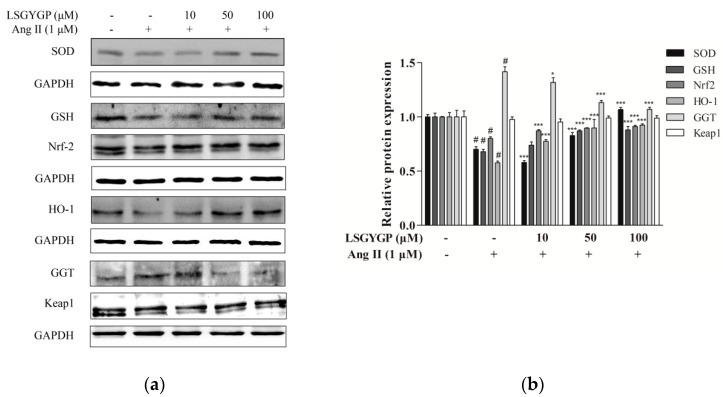
Effects of LSGYGP on the expression of superoxide dismutase (SOD), gamma-glutamyltransferase (GGT), glutathione (GSH), and nuclear erythroid 2-related factor 2 (Nrf2) pathway in HUVEC (**a**,**b**). Protein expression was by western blot, glyceraldehyde-3-phosphate dehydrogenase (GAPDH) was as control. ^#^
*p* < 0.001, compared with blank group (untreated cells); * *p* < 0.05, *** *p* < 0.001, compared with control group (Ang II-treated cells).

**Figure 4 marinedrugs-17-00431-f004:**
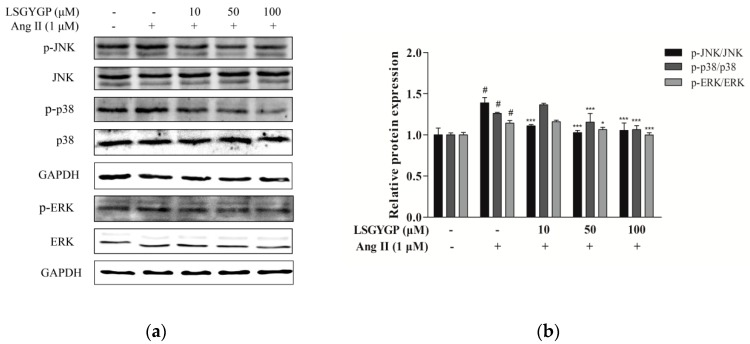
(**a**) HUVEC were exposed to different concentrations (10, 50, and 100 μM) of LSGYGP with 1 μM Ang II to stimulated mitogen-activated protein kinases (MAPK) phosphorylation in cells. (**b**) Equal amounts of protein were loaded in each lane. # indicate significantly compared with untreated cells; ^#^
*p* < 0.001, compared with blank group (untreated cells); * *p* < 0.05, *** *p* < 0.001, compared with control group (Ang II-treated cells).

**Figure 5 marinedrugs-17-00431-f005:**
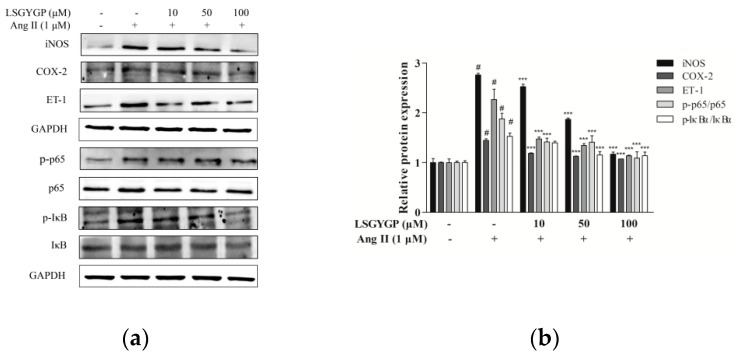
Effects of LSGYGP on the expression of iNOS, COX-2, NF-κB p65, and IκB*α* in HUVEC (**a**,**b**). Protein expression was by western blot, GAPDH, NF-κB p65, and IκB*α* were as control, respectively. (**c**) The effect of LSGYGP on translocation of NF-κB p65 in HUVEC. Cells were pre-treated with LSGYGP (10 and 100 μM) and subsequently treated with Ang II (1 μM) for 24 h. Nucleus was stained with DAPI and NF-κB p65 was immunostained with p65 antibody. (**d**,**e**) LSGYGP suppressed the NF-κB activity inside the nucleus of NF-κB in Ang II-stimulated HUVEC. Electrophoretic mobility shift assay (EMSA) was performed to determine the NF-κB activity in nuclear reaction by using DNA probe specific to NF-κB p65. ^#^
*p* < 0.001, compared with blank group (untreated cells); ** *p* < 0.01, *** *p* < 0.001, compared with control group (Ang II-treated cells).

**Figure 6 marinedrugs-17-00431-f006:**
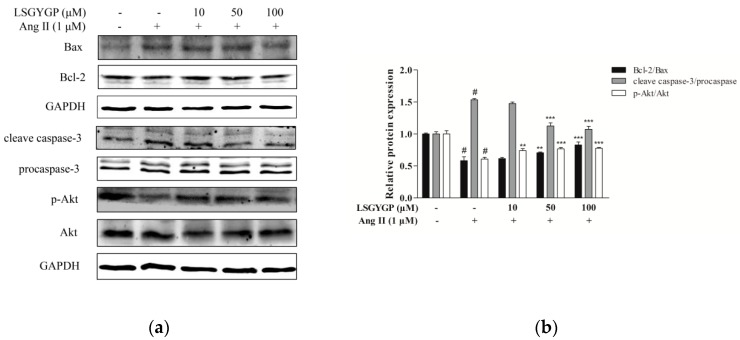
(**a**) The expressions of Bax, Bcl-2, procaspase-3, caspase-3 (p20), and phosphorylation of Akt in HUVEC. GAPDH was used as an internal control. (**b**) The ratios of Bcl-2/Bax and cleaved-caspase-3/procaspase-3 were calculated. ^#^
*p* < 0.001, compared with blank group (untreated cells); ** *p* < 0.01, *** *p* < 0.001, compared with control group (Ang II-treated cells).

**Figure 7 marinedrugs-17-00431-f007:**
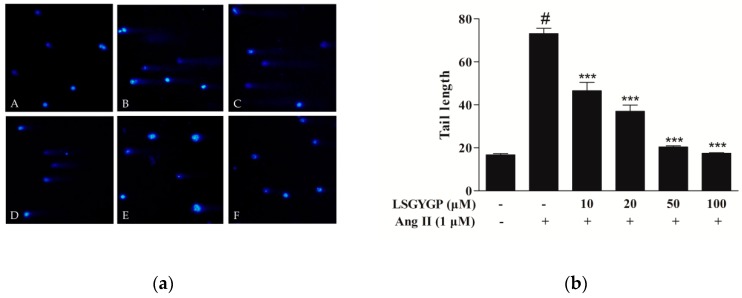
(**a**) Comet assay of HUVEC: (A) cells without treatment (the blank group); (B) cells exposed to 1 μM Ang II (the control group); (C, D, E, and F) cells pretreated with LSGYGP (10, 20, 50, and 100 µM, respectively) prior to treatment with 1 μM Ang II, all both followed by staining with DAPI. Images were obtained using an inverted fluorescence microscope with blue fluorescence (magnification: 10×). (**b**) Tail lengths of the comets were analyzed. ^#^
*p* < 0.001, compared with blank group (untreated cells); *** *p* < 0.001, compared with control group (Ang II-treated cells).

**Figure 8 marinedrugs-17-00431-f008:**
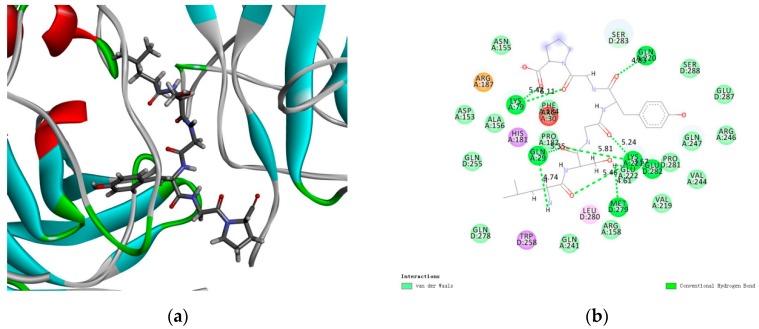
General and local overview poses of the best interaction poses after automated docking of peptide-NF-κB p65 active site (**a**), amino acids of LSGYGP involved in hydrogen bonds are represented by thin sticks; (**b**) Bi-dimensional (2D) diagrams of predicted interactions between ligand and NF-κB p65 amino acid residues. LSGYGP is draw by gray lines, and hydrogen bonds of them shown with green dashed lines.

**Figure 9 marinedrugs-17-00431-f009:**
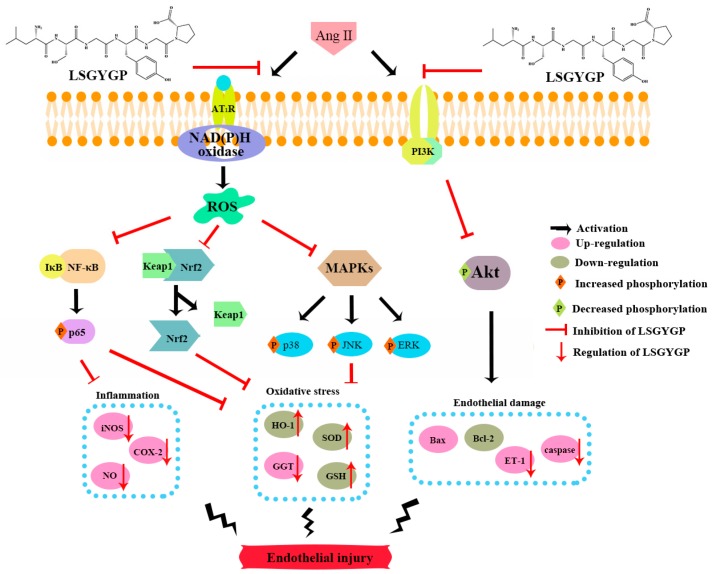
Summary diagram of signaling pathway of this study.

**Table 1 marinedrugs-17-00431-t001:** LSGYGP contacts with NF-κB (PDB: 1IKN).

Number	Interacting Atoms	Distance (Å)	Interaction Force
1	Lys79	5.42	Conventional Hydrogen Bond
2		6.11	
3	Gln220	4.83	
4	Gln29	5.35	
5		4.74	
6	Met279	4.61	
7	Glu282	3.57	
8	Lys221	5.24	
9		5.81	
10		5.46	

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
