# Peer review of "In Vitro Vascular-Protective Effects of a Tilapia By-Product Oligopeptide on Angiotensin II-Induced Hypertensive Endothelial Injury in HUVEC by Nrf2/NF-κB Pathways"

_marinedrugs, 2019, doi:10.3390/md17070431_

Reviewer 1 Report

The authors investigate the role of tilapia skin-derived LSGYGP on Ang II-induced endothelial damage. They show that LSGYGP protects HUVEC from AngII induced cell death, reverse AngII induced oxidative stress and endothelial dysfunction. The manuscript is well prepared. One question I have with the manuscript is that although the authors show the molecular docking of LSGYGP, it is not clear what is the target of this oligopeptide if the oligopeptide is cell permeable or it binds any cell surface receptor.  

Author Response

Response to Reviewer 1

Dear reviewer,

On behalf of my co-authors, we thank you very much for giving us an opportunity to revise our manuscript and we appreciate editor very much for your positive and constructive comments and suggestions on our manuscript entitled In Vitro Vascular-protective Effects of a Tilapia By-product Oligopeptide on Angiotensin II-induced Hypertensive Endothelial Injury in HUVEC by Nrf2/NF-κB Pathways(ID: Marine Drugs-530555).

Firstly, our manuscript has undergone English language editing by MDPI (ID: 10562).

We have studied editor’s comments carefully and have made revision in the manuscript. We have tried our best to revise our manuscript according to the comments.

Comments

Although the authors show the molecular docking of LSGYGP, it is not clear what is the target of this oligopeptide if the oligopeptide is cell permeable or it binds any cell surface receptor.

Response: Thanks for your comment. Based on your suggestion, we have consulted and collected relevant reference, we will discuss this question with two cases as follows.

Firstly, if oligopeptide LSGYGP is cell permeable, it would get into cell through peptide transport 1 (PepT1)-mediated route, paracellular route via tight junctions (TJs), transcytosis route or passive transcellular diffusion[1–3]. Due to LSGYGP is a six amino acid oligopeptide with a molecular weight of 592.26 Da, on the background of digestive stability it does not conform to the transport mode of PepT1 that drives small peptides (di- and tripeptides).

Therefore, we speculate that it is 1) paracellular route via TJs that peptides could be transported across cell monolayers. Amino acid composition of our peptide is partial similar with those of some food-derived peptides which transported by the energy-independent paracellular route via TJs, such as Arg-Leu-Ser-Phe-Asn-Pro[4], Lys-Val-Leu-Pro-Val-Pro[5]. If our peptide get into cell through this nondigestive transport, paracellular route, could avoid the hydrolysis of intracellular peptidases[6], so intact peptide is possibly effect on HUVEC. 2) Transcytosis, is an energy-dependent transcellular pathway that involves apical endocytic uptake via internalisation, transcytotic vesicle transport and basolateral secretion. It requires peptides with high hydrophobicity for interact with the apical lipid surface via hydrophobic interactions. Our peptide possess hydrophobic amino acid (Leu and Pro) with the N/C-terminal, but further study is using wortmannin, an inhibitor of phosphoinositide 3-kinase, to identify. 3) Simple passive transcellular diffusion. A number of peptides are transported across the intestinal brush-border membrane via simple passive transcellular diffusion. Highly hydrophobic peptides tend to be transported by simple passive transcellular diffusion. This pathway of peptide transport further be quantify via passive diffusion regulators. We speculate that the paracellular route via TJs is how the peptide is most likely to enter the cell while above transport pathways both were further evaluated via the Caco-2 cell monolayer model.

Another, oligopeptide is binding with cell receptor.

1) Angiotensin type 1 (AT1) receptor. Angiotensin type 1 receptor (AT1R), the most prevalent and best characterized of the octapeptide Ang II (Asp-Arg-Val-Tyr-Ile-His-Pro-Phe) receptors, is a 7-transmembrane, G protein-coupled receptor (GPCR) with a short C-terminal cytoplasmic domain that is important for intracellular trafficking and targeting of the receptor, and also for activation of ligand-induced signaling pathways[7]. The actions of Ang II include binding to the AT1R, thereby activating membrane-bound NAD(P)H oxidase for the formation of reactive oxygen species (ROS), particularly superoxide anion[8]. We can identify LSGYGP bind with AT1R through determinate the location for GABARAP, a small (117 aa) trafficking protein, the juxtaposition of the binding domain next to the cellular membrane would appear to be an ideal location for GABARAP-a protein with a propensity to covalently attach to membrane lipids-in carrying out its vesicular trafficking function[9].

2) Endothelial cell protein C receptor (EPCR). The sequence of EPCR predicts it to be a type 1 trans membrane glycoprotein, a survey of cultured cells indicated that message and binding function was endothelial cell-specific. Oligopeptide is possibly binding with EPCR through testimony of significant loss of protein C/activated protein C (APC) binding affinity[10].

We have compiled above information in red revision to “Discussion” part of our manuscript for discussion and outlook.

“Especially, what is worthy to study that whether Ang II binding to the angiotensin type 1-receptor (AT1R) and activating membrane-bound NAD(P)H oxidase for the formation of ROS or oligopeptide LSGYGP bind with AT1R through determinate the location for GABARAP. Besides, bioavailability and transport mechanisms of oligopeptide are remains to research for following bioactivity design and application of peptide drugs, these problems need to be overcome by using different design strategies in the future.”

1.    Xu, Q.; Yan, X.; Zhang, Y.; Wu, J. Current understanding of transport and bioavailability of bioactive peptides derived from dairy proteins: a review. International Journal of Food Science & Technology 2019, 54, 1930-1941.

2.    Martin, M.; Deussen, A. Effects of natural peptides from food proteins on angiotensin converting enzyme activity and hypertension. Critical Reviews in Food Science and Nutrition 2019, 59, 1264-1283.

3.    Jao, C.-L.; Huang, S.-L.; Hsu, K.-C. Angiotensin I-converting enzyme inhibitory peptides: Inhibition mode, bioavailability, and antihypertensive effects. BioMedicine 2012, 2, 130-136.

4.    Guo, Y.; Gan, J.; Zhu, Q.; Zeng, X.; Sun, Y.; Wu, Z.; Pan, D. Transepithelial transport of milk-derived angiotensin I-converting enzyme inhibitory peptide with the RLSFNP sequence: Transepithelial transport of ACE-inhibitory peptide RLSFNP. Journal of the Science of Food and Agriculture 2018, 98, 976-983.

5.    Sun, H.; Liu, D.; Li, S.; Qin, Z. Transepithelial Transport Characteristics of the Antihypertensive Peptide, Lys-Val-Leu-Pro-Val-Pro, in Human Intestinal Caco-2 Cell Monolayers. Bioscience, Biotechnology, and Biochemistry 2009, 73, 293-298.

6.    Shimizu, M.; Son, D.O. Food-Derived Peptides and Intestinal Functions. Current Pharmaceutical Design 2007, 13, 885-895.

7.    Becher, U.M.; Endtmann, C.; Tiyerili, V.; Nickenig, G.; Werner, N. Endothelial Damage and Regeneration: The Role of the Renin-Angiotensin-Aldosterone System. Current Hypertension Reports 2011, 13, 86-92.

8.    Zhang, H. Angiotensin II-induced superoxide anion generation in human vascular endothelial cells Role of membrane-bound NADH-/NADPH-oxidases. Cardiovascular Research 1999, 44, 215-222.

9.    Alam, J.; DeHaro, D.; Redding, K.M.; Re, R.N.; Cook, J.L. C-terminal processing of GABARAP is not required for trafficking of the angiotensin II type 1A receptor. Regulatory Peptides 2010, 159, 78-86.

10.  Gleeson, E.M.; O’Donnell, J.S.; Preston, R.J.S. The endothelial cell protein C receptor: cell surface conductor of cytoprotective coagulation factor signaling. Cellular and Molecular Life Sciences 2012, 69, 717-726.

We have studied reviewer’s comments carefully and have made revision which marked in red in the revised manuscript. We have tried our best to revise our manuscript according to the comments. Attached please find the revised version, which we would like to submit for your kind consideration.

We would like to express our great appreciation to you and reviewers for comments on our paper. Looking forward to hearing from you.

Thank you and best regards.

Yours sincerely

Zhong-Ji Qian, Ph.D.

Professor

Shenzhen Institute of Guangdong Ocean University, Shenzhen 518108, China

School of Chemistry and Environment, Guangdong Ocean University, Zhanjiang 524088, China

Tel: +88-759-2396270

Reviewer 2 Report

The Authors in this paper aim to investigate the oligopeptide LSGYGP and its protective affect on oxidative stress and endothelial injury after Ang-II stimulation.

The following items need to be addressed:

In figure 1 (figure legend) and section 2.1 you refer to part b as the cytotoxicity of LSGYGP in HUVEC. I think you mean the cytoprotection of LSGYGP. Section 2.2 is directly repeated from section 2.1. The images in Figure 2a do not seem to reflect the graph in 2b specifically the 50uM LSGYGP panel.

The Western blots in figure 3a, 4a, 5a and 6a: Is the blot the same blot used for all antibodies? Are these re-probed? If not can you please provide the GAPDH for each blot. The bands for the p38 and p-p38 do not match in terms of size or shape which indicates that these may not be the same blot, this is the case for many of the phospho detected blots. All westerns are too cropped can you please provide more of the blot, preferably the entire blot. For the western in figure 6a can you please show more of the pro-caspase 3 staining as there appears to be two bands one of which is partially cut off.

In Figure 5c it is difficult to determine if the p65 in the images is in the nucleus unless you use confocal with z stack. The EMSA assay is not very convincing, you should consider using a p65 sensitive promoter reporter gene assay to look at p65 activity to give a clearer indication of activation of NFkB-p65 and whether LSGYGP blocks this activation.

Figure 7a. Can you please enlarge these images.

If the LSGYGP is binding to NFkB-p65 then you would expect that IP1/IP3 would not be affected. It might be useful to show this using an IP1/IP3 assay. Additionally, if you used siRNA to p65 you would anticipate the same affect as using LSGYG.

Author Response

Response to Reviewer 2

Dear reviewer,

On behalf of my co-authors, we thank you very much for giving us an opportunity to revise our manuscript and we appreciate editor very much for your positive and constructive comments and suggestions on our manuscript entitled “In Vitro Vascular-protective Effects of a Tilapia By-product Oligopeptide on Angiotensin II-induced Hypertensive Endothelial Injury in HUVEC by Nrf2/NF-κB Pathways” (ID: Marine Drugs- 530555).

Firstly, our manuscript has undergone English language editing by MDPI (ID: 10562).

We have studied editor’s comments carefully and have made revision in the manuscript. We have tried our best to revise our manuscript according to the comments.

Comments

Point 1: In figure 1 (figure legend) and section 2.1 you refer to part b as the cytotoxicity of LSGYGP in HUVEC. I think you mean the cytoprotection of LSGYGP.

Response 1: Thanks for your careful comment. We have corrected this with red in the revision version. Figure 1 (figure legend) “Cytotoxicity of LSGYGP in HUVEC (b) and of LSGYGP with Ang II-treated cells (c).” corrected to “Effect of LSGYGP on the viability of HUVEC (b) and protective effect of LSGYGP on the viability of Ang II-treated HUVEC (c).”

Point 2: Section 2.2 is directly repeated from section 2.1.

Response 2: Thanks for your careful work. We have corrected this mistake with red in the revision version.

Corrected Section 2.2: “As Figure 2a shows, the Ang II group had an obvious effect on NO and ROS levels, as compared to the untreated control. Remarkably, 10 μM treatment with LSGYGP decreased the Ang II-induced increase in NO and ROS generation. LSGYGP significantly attenuates NO and ROS production in a dose-dependent manner.”

Point 3: The images in Figure 2a do not seem to reflect the graph in 2b specifically the 50 uM LSGYGP panel.

Response 3: Thanks for your comment. 50 μM LSGYGP have a significant inhibition to attenuate NO and ROS generations in HUVEC, these inhibited effects compared with untreated-group is significant as shown in the difference analysis diagram below.

Point 4: The Western blots in figure 3a, 4a, 5a and 6a: Is the blot the same blot used for all antibodies? Are these re-probed? If not can you please provide the GAPDH for each blot. The bands for the p38 and p-p38 do not match in terms of size or shape which indicates that these may not be the same blot, this is the case for many of the phospho detected blots. All westerns are too cropped can you please provide more of the blot, preferably the entire blot. For the western in figure 6a can you please show more of the pro-caspase 3 staining as there appears to be two bands one of which is partially cut off.

Response 4: Thanks for your comment. Some strips are not the same strip of protein that comes out of the same electrophoresis. We have already sorted them out and added the corresponding GAPDH in Figures. Blots of p38 and p-p38 were comes from different electrophoresis results before, they were electrophored and photographed again as following.

We have widened pro-caspase 3 staining which show two bands, to reveal its more details as shown in Figure 6a, and all the other strips are fully displayed as same as we can.

Please understand, for reason that number of western blots is a little more, we have tried our best to provide more bolt for picture combination. Entire blots were provided at the end of this respond.

Point 5: In Figure 5c it is difficult to determine if the p65 in the images is in the nucleus unless you use confocal with z stack. The EMSA assay is not very convincing, you should consider using a p65 sensitive promoter reporter gene assay to look at p65 activity to give a clearer indication of activation of NF-κB p65 and whether LSGYGP blocks this activation.

Response 5: Thanks for your careful work. Based on your suggestion, we have re-executed the EMSA experiment based on increasement of Streptavidin-HRP Conjugate and carry out differential analysis. As you can see in Figure 5d and 5e, nucleus protein was tightly bind with NF-κB p65 probe, and LSGYGP blocks activation of NF-κB p65.

Please understand that for the limitations of experimental equipment, we can't use the confocal microscope to take high-resolution experimental pictures. As for you mentioned p65 sensitive promoter reporter gene assay, we are sorry about unable to provide this result for sample and time reasons. We indeed access to some reference about sensitive promoter reporter gene assay, which deserve be used as an auxiliary certificate to the research. The relative luciferase activity was assayed using the Promega Dual-Luciferase reporter assay system in Zhu’study [1]. The reporter gene activities in Duan’s study were assayed using the Dual-Light Combined Reporter Gene Assay System for detection of luciferase and h-galactosidase [2], which deeeply shown the activation of nuclear factor-κB in protein and gene, this study provide good strategy for our further study.

1.    Zhu, N.; Zhang, D.; Chen, S.; Liu, X.; Lin, L.; Huang, X.; Guo, Z.; Liu, J.; Wang, Y.; Yuan, W.; et al. Endothelial enriched microRNAs regulate angiotensin II-induced endothelial inflammation and migration. Atherosclerosis 2011, 215, 286–293.

2.    Duan, J.; Friedman, J.; Nottingham, L.; Chen, Z.; Ara, G.; Van Waes, C. Nuclear factor- B p65 small interfering RNA or proteasome inhibitor bortezomib sensitizes head and neck squamous cell carcinomas to classic histone deacetylase inhibitors and novel histone deacetylase inhibitor PXD101. Mol. Cancer Ther. 2007, 6, 37–50.

Point 6: Figure 7a. Can you please enlarge these images.

Response 6: Thanks for your comment. Based on your suggestion, we have enlarged Figure 7a as we can and revised figure legend with red as follow.

Point 7: If the LSGYGP is binding to NF-κB p65 then you would expect that IP1/IP3 would not be affected. It might be useful to show this using an IP1/IP3 assay. Additionally, if you used siRNA to p65 you would anticipate the same affect as using LSGYGP.

Response 7: Thanks for your suggestion. According to your suggestion, we indeed access to some reference about IP1/IP3 assay and siRNA, which deserve be used as a follow-up to the research. We guess you mentioned IP1/IP3 assay is an equilibrium calcium assay that is able to determine the correct (i.e. internally consistent) pharmacological profiles of all tested compounds. We can determine whether LSGYGP is binding to NF-κB p65 by calcium release from IP3-sensitive endoplasmic reticulum in further study. Besides, we have consulted some reference and found there are some active ingredient or compound could suppress expression of p65 via siRNA, such as resveratrol [3], Sauchinone [4], and Galectin‐3 [5], but they are different in target of siRNA. Please understand that for the limitations of experimental equipment, sample and time, we are sorry about unable to provide this result.

3.    Pan, W.; Yu, H.; Huang, S.; Zhu, P. Resveratrol Protects against TNF-α-Induced Injury in Human Umbilical Endothelial Cells through Promoting Sirtuin-1-Induced Repression of NF-KB and p38 MAPK. PLOS ONE 2016, 11, e0147034.

4.    Li, B.; Lee, Y.J.; Kim, Y.C.; Yoon, J.J.; Lee, S.M.; Lee, Y.P.; Kang, D.G.; Lee, H.S. Sauchinone from Saururus chinensis protects vascular inflammation by heme oxygenase-1 induction in human umbilical vein endothelial cells. Phytomedicine 2014, 21, 101–108.

5. Chen, X.; Lin, J.; Hu, T.; Ren, Z.; Li, L.; Hameed, I.; Zhang, X.; Men, C.; Guo, Y.; Xu, D.; et al. Galectin‐3 exacerbates ox‐LDL‐mediated endothelial injury by inducing inflammation via integrin β1‐RhoA‐JNK signaling activation. J. Cell. Physiol. 2019, 234, 10990–11000.

Entire blots of western blotting as following:

We have studied reviewer’s comments carefully and have made revision which marked in red in the revised manuscript. We have tried our best to revise our manuscript according to the comments. Attached please find the revised version, which we would like to submit for your kind consideration.

We would like to express our great appreciation to you and reviewers for comments on our paper. Looking forward to hearing from you.

Thank you and best regards.

Yours sincerely

Zhong-Ji Qian, Ph.D.

Professor

Shenzhen Institute of Guangdong Ocean University, Shenzhen 518108, China

School of Chemistry and Environment, Guangdong Ocean University, Zhanjiang 524088, China

Tel: +88-759-2396270

Reviewer 3 Report

In this study, LSGYGP was evaluated its protective effect on oxidative stress and endothelial injury in Angiotensin II-stimulated human umbilical vein endothelial cells (HUVEC). These results suggested this oligopeptide is effectively to protect Ang II-induced HUVEC injury through reducing oxidative stress and alleviating endothelial damage. I recommend accepting this paper to be published after addressing the following issues.

1. The language should be well polished, and many grammatical mistakes need to be corrected.

2. In Figure 1(a), the chemical structure of LSGYGP should be drawn by ChemDraw software in a correct format. The exact configurations of each chiral carbon also need to be shown.

3. In Figure 1(c), for the control group, was 1 μM AngII used in the test? This important information must be listed in the title of the figure.

4. In the manuscript, lots of assays were performed to try to prove that LSGYGP displayed effective activity as potential agents to treat cardiovascular disease. To summarize the results, as well as to illustrate the internal connection, a figure is suggested to be prepared to show the mechanism among LSGYGP with AngII, iNOS, ROS, COX-2, NF-κB, SOD, GSH, MAPK, HO-1, Nrf2 and Akt.

Author Response

Response to Reviewer 3

Dear reviewer,

On behalf of my co-authors, we thank you very much for giving us an opportunity to revise our manuscript and we appreciate editor very much for your positive and constructive comments and suggestions on our manuscript entitled In Vitro Vascular-protective Effects of a Tilapia By-product Oligopeptide on Angiotensin II-induced Hypertensive Endothelial Injury in HUVEC by Nrf2/NF-κB Pathways” (ID: Marine Drugs- 530555).

Firstly, our manuscript has undergone English language editing by MDPI (ID: 10562).

We have studied editor’s comments carefully and have made revision in the manuscript. We have tried our best to revise our manuscript according to the comments.

Comments

Point 1: The language should be well polished, and many grammatical mistakes need to be corrected.

Response 1: Thanks for your kind comment. Our manuscript has undergone English language editing by MDPI (ID: 10562). The text has been checked for correct use of grammar and common technical terms, and edited to a level suitable for reporting research in a scholarly journal.

Point 2: In Figure 1(a), the chemical structure of LSGYGP should be drawn by ChemDraw software in a correct format. The exact configurations of each chiral carbon also need to be shown.

Response 2: Thanks for your careful work. We have correct Figure 1(a) in the revision version as follow. Exact configurations of each chiral carbon were marked with arrows in Figure 1(a).

Point 3: In Figure 1(c), for the control group, was 1 μM Ang II used in the test? This important information must be listed in the title of the figure.

Response 3: Thanks for your careful comment. 1 μM Ang II indeed was used in all groups except blank group. Based on your comment, we added this information to Figure 1(c) in the revision version as follow, and we have also added this all the figures in manuscript.

Point 4: To summarize the results, as well as to illustrate the internal connection, a figure is suggested to be prepared to show the mechanism among LSGYGP with AngII, iNOS, ROS, COX-2, NF-κB, SOD, GSH, MAPK, HO-1, Nrf2 and Akt.

Response 4: Thanks for your comment. We added this suggestion to Figure 9 in the revision version, the mechanism among LSGYGP with relevant proteins and pathways was marked with red symbol as follow.

We have studied reviewer’s comments carefully and have made revision which marked in red in the revised manuscript. We have tried our best to revise our manuscript according to the comments. Attached please find the revised version, which we would like to submit for your kind consideration.

We would like to express our great appreciation to you and reviewers for comments on our paper. Looking forward to hearing from you.

Thank you and best regards.

Yours sincerely

Zhong-Ji Qian, Ph.D.

Professor

Shenzhen Institute of Guangdong Ocean University, Shenzhen 518108, China

School of Chemistry and Environment, Guangdong Ocean University, Zhanjiang 524088, China

Tel: +88-759-2396270

Reviewer 4 Report

The study investigates the role of lsgygp on ang II induced oxidative stress and injury in EC. The peptide reduced AngII induced death, NO and ROS production. This was paralleled by increased SOD, GSH and NRF2. JNK and p38MAPK activation were reversed as was increased expression of iNOS, ET1 and NFkB. Increased apoptosis and markers for apoptosis were also reversed. A model is proposed in which the peptide blocks numerous aspects of angII induced EC damage. The peptide has previously been shown to protect against UV induced ROS-generation and damage.

Whereas the data on cell death and NO and ROS generation and NFkB translocation are convincing, the western blotting data are not. The quantification is based on triplicates showing tiny effects with very small error bars. When looking at the low quality of the blots, it is impossible to imagine that the data reflect reality. The GAPDH blots are overexposed so one can’t assess minor differences in loading between the lanes. I’m most upset by the data on GSH, Nrf2, HO1, pJNK, ET1 and NFkB activation by EMSA. These data cannot be printed.

Author Response

Response to Reviewer 4

Dear reviewer,

On behalf of my co-authors, we thank you very much for giving us an opportunity to revise our manuscript and we appreciate editor very much for your positive and constructive comments and suggestions on our manuscript entitled “In Vitro Vascular-protective Effects of a Tilapia By-product Oligopeptide on Angiotensin II-induced Hypertensive Endothelial Injury in HUVEC by Nrf2/NF-κB Pathways” (ID: Marine Drugs- 530555).

Firstly, our manuscript has undergone English language editing by MDPI (ID: 10562).

We have studied editor’s comments carefully and have made revision in the manuscript. We have tried our best to revise our manuscript according to the comments.

Comments

Point 1: The quantification is based on triplicates showing tiny effects with very small error bars. When looking at the low quality of the blots, it is impossible to imagine that the data reflect reality. The GAPDH blots are overexposed so one can’t assess minor differences in loading between the lanes. I’m most upset by the data on GSH, Nrf2, HO-1, p-JNK, ET-1 and NF-κB activation by EMSA.

Response 1: Thanks for your comment. Based on your suggestion, we re-do experiments with those weak expression bands, increasing the concentration of their primary antibodies and reducing GAPDH’s concentration of primary antibodies. We analyzed the differences in all GAPDH and found that they showed no significant difference on triplicates as following.

Proteins including GSH, Nrf2, HO-1, p-JNK, ET-1 and NF-κB activation by EMSA were electrophored and photographed again as following. We have corrected these pictures in the revision version as following.

Western blots of Fig 3a: GSH, Nrf2, HO-1 and GAPDH

Western blots of Fig 4a: p-JNK, JNK, p-p38, p38 and GAPDH

         Western blots of Fig 5a: ET-1 and GAPDH 

EMSA of Fig 5d, 5e: NF-κB p65

We have studied reviewer’s comments carefully and have made revision which marked in red in the revised manuscript. We have tried our best to revise our manuscript according to the comments. Attached please find the revised version, which we would like to submit for your kind consideration.

We would like to express our great appreciation to you and reviewers for comments on our paper. Looking forward to hearing from you.

Thank you and best regards.

Yours sincerely

Zhong-Ji Qian, Ph.D.

Professor

Shenzhen Institute of Guangdong Ocean University, Shenzhen 518108, China

School of Chemistry and Environment, Guangdong Ocean University, Zhanjiang 524088, China

Tel: +88-759-2396270

Round  2

Reviewer 2 Report

The Authors examined the ability of the LSGYGP peptide from Tilapia to prevent the effects of AngII in endothelial cells. Here the Authors showed that LSGYGP was able to prevent oxidative stress and endothelial disfunction by suppressing the NFkB pathway and up regulating the NRF-2 pathway using primarily protein data via western blots.

Overall Comments

The strengths of the paper are the use of protein data to demonstrate the changes seen after AngII treatment which was then reversed using the peptide. The weaknesses of the study is that the Authors have not examined how the peptide prevents AngII from exerting its effects, though the Authors acknowledge this and is in the discussion as a future study. 

I would like to thank the Authors for providing the full blots so that I could review the images more closely and for adding in GAPDH for each blot used. The Authors addressed most of my concerns. 

Please address the following:

The results in figure 5d & e are much clearer. However, Figure 5c still cannot show that the protein has entered the nucleus, I believe that p65 might have entered the nucleus based on 5d &e but unfortunately without confocal images the text that describes figure 5c in the results (lines 146-151) would not be accurate. If the wording was changed to suggests that p65 might be transported to the nucleus then that would be better. I understand that not all labs have access to a confocal microscope.  

The diagram in figure 9 should also include the Angiotensin receptors. 

Minor remarks

Please address the following issues in the text (please address the bold text):

Line 62: Moreover, its induction, upon undergone oxidative stress

Line 75: the process of tilapia filleting other parts of the fish was wasted, for example the frame and skin

Line 83: investigating whether it is able to protect against

Line 142: expression (not expressions)

Line 143: markedly blocked the AngII induced changes.

Lines 281-286: This paragraph has poor English and typos. 

Author Response

Round 2- Response to Reviewer 2

Dear reviewer,

On behalf of my co-authors, we thank you very much for giving us an opportunity to revise our manuscript and we appreciate editor very much for your positive and constructive comments and suggestions on our manuscript entitled “In Vitro Vascular-protective Effects of a Tilapia By-product Oligopeptide on Angiotensin II-induced Hypertensive Endothelial Injury in HUVEC by Nrf2/NF-κB Pathways” (ID: Marine Drugs- 530555).

We have studied editor’s comments carefully and have made revision in the manuscript. We have tried our best to revise our manuscript according to the comments.

Comments

Point 1: The results in figure 5d & e are much clearer. However, Figure 5c still cannot show that the protein has entered the nucleus, I believe that p65 might have entered the nucleus based on 5d &e but unfortunately without confocal images the text that describes figure 5c in the results (lines 146-151) would not be accurate. If the wording was changed to suggests that p65 might be transported to the nucleus then that would be better. I understand that not all labs have access to a confocal microscope.

Response 1: Thanks for your kindly undertanding. Based on your suggestion, the text that describes Figure 5c and Figure 5d in the results have corrected toImmunocytochemistry and EMSA were applied to further investigate the translocation of NF-κB p65 in Ang II-stimulated HUVEC. Observing images in Figure 5c, the NF-κB p65 sub-unit was transported to the nucleus after Ang II stimulation, as indicated by the NF-κB p65 protein level in the western blot test. Employing image analysis, treatment with Ang II presented a significant increase in the DNA-binding activity of NF-κB, whereas treatment with LSGYGP significantly reduced the Ang II-induced DNA-binding activity of NF-κB (Figure 5d, 5e), this revealed that p65 might enter the nucleus based on DNA-binding activity. with red in the revision version.

Point 2: The diagram in figure 9 should also include the Angiotensin receptors.

Response 2: Thanks for your comment. We have added this suggestion to Figure 9 in the revision version.

Figure 9

Point 3: Minor remarks, please address the following issues in the text (please address the bold text). Line 62: Moreover, its induction, upon undergone oxidative stress; Line 75: the process of tilapia filleting other parts of the fish was wasted, for example the frame and skin; Line 83: investigating whether it is able to protect against; Line 142: expression (not expressions); Line 143: markedly blocked the AngII induced changes; Lines 281-286: This paragraph has poor English and typos.

Response 3: Thanks for your comment. We have corrected these mistakes with red in the revision version.

Corrected Line 62: Moreover, under induction of oxidative stress and nitric oxide, it is mainly regulated by the activation of Nrf-2 and MAPK/ERK pathways in vascular and endothelial cells.

Corrected Line 75: It’s reported that during the process of tilapia filleting other parts of the fish was wasted, for example the frame and skin

Corrected Line 83: This study is aimed to investigate whether it is able to protect against cardiovascular injury.

Corrected Line 142: As depicted in Figure 5b, expression of COX-2 and ET-1 mainly increased after exposure to Ang II,

Corrected Line 143: but treatment of 10 μM LSGYGP markedly blocked the AngII induced changes.

Corrected Lines 281-286: “Future research should further discuss and confirm that how Ang II to binds the angiotensin type 1-receptor (AT1R) and activates membrane-bound NAD(P)H oxidase for the formation of ROS; how oligopeptide LSGYGP to binds with AT1R through determinate the location for GABARAP. Besides, bioavailability and transport mechanisms of oligopeptide are remains to research for following bioactivity design and application of peptide drugs, these are the key concerns attempted to overcome by using different design strategies in the future.

We have studied reviewer’s comments carefully and have made revision which marked in red in the revised manuscript. We have tried our best to revise our manuscript according to the comments. Attached please find the revised version, which we would like to submit for your kind consideration.

We would like to express our great appreciation to you and reviewers for comments on our paper. Looking forward to hearing from you.

Thank you and best regards.

Yours sincerely

Zhong-Ji Qian, Ph.D.

Professor

Shenzhen Institute of Guangdong Ocean University, Shenzhen 518108, China

School of Chemistry and Environment, Guangdong Ocean University, Zhanjiang 524088, China

Tel: +88-759-2396270

Reviewer 3 Report

The authors have addressed most of the concerns. I recommend accepting this manuscript in present form.

Author Response

Round 2- Response to Reviewer 3

Dear reviewer,

On behalf of my co-authors, we thank you very much for giving us an opportunity to revise our manuscript and we appreciate editor very much for your positive and constructive comments and suggestions on our manuscript entitled In Vitro Vascular-protective Effects of a Tilapia By-product Oligopeptide on Angiotensin II-induced Hypertensive Endothelial Injury in HUVEC by Nrf2/NF-κB Pathways” (ID: Marine Drugs- 530555).

We have studied editor’s comments carefully and have made revision in the manuscript. We have tried our best to revise our manuscript according to the comments.

Comments

The authors have addressed most of the concerns. I recommend accepting this manuscript in present form.

Response: Thanks for your comment. We are very grateful for your valuable comments and suggestions.

We have studied reviewer’s comments carefully and have made revision which marked in red in the revised manuscript. We have tried our best to revise our manuscript according to the comments. Attached please find the revised version, which we would like to submit for your kind consideration.

We would like to express our great appreciation to you and reviewers for comments on our paper. Looking forward to hearing from you.

Thank you and best regards.

Yours sincerely

Zhong-Ji Qian, Ph.D.

Professor

Shenzhen Institute of Guangdong Ocean University, Shenzhen 518108, China

School of Chemistry and Environment, Guangdong Ocean University, Zhanjiang 524088, China

Tel: +88-759-2396270

Reviewer 4 Report

Blots have improved. There is still need for linguistic revision. Here are two examples of problematic English:

"Especially, what is worthy to study that whether Ang II binding to the

282  angiotensin type 1-receptor (AT1R) and activating membrane-bound NAD(P)H oxidase for the

283  formation of ROS, what is worthy to study tthat whether or oligopeptide LSGYGP bind with AT1R

284  through determinate the location for GABARAP. Besides, bioavailability and transport mechanisms

285  of oligopeptide are remains to research for following bioactivity design and application of peptide

286  drugs., these problems need to be overcome by using different design strategies in the future."

"

# indicate significantly compared with blank group (untreated cells);

"

Author Response

Round 2- Response to Reviewer 4

Dear reviewer,

On behalf of my co-authors, we thank you very much for giving us an opportunity to revise our manuscript and we appreciate editor very much for your positive and constructive comments and suggestions on our manuscript entitled “In Vitro Vascular-protective Effects of a Tilapia By-product Oligopeptide on Angiotensin II-induced Hypertensive Endothelial Injury in HUVEC by Nrf2/NF-κB Pathways” (ID: Marine Drugs- 530555).

We have studied editor’s comments carefully and have made revision in the manuscript. We have tried our best to revise our manuscript according to the comments.

Comments

Point 1: Here are two examples of problematic English:

"Especially, what is worthy to study that whether Ang II binding to the 282  angiotensin type 1-receptor (AT1R) and activating membrane-bound NAD(P)H oxidase for the 283  formation of ROS, what is worthy to study tthat whether or oligopeptide LSGYGP bind with AT1R 284  through determinate the location for GABARAP. Besides, bioavailability and transport mechanisms 285  of oligopeptide are remains to research for following bioactivity design and application of peptide 286  drugs, these problems need to be overcome by using different design strategies in the future."

# indicate significantly compared with blank group (untreated cells);

Response 1: Thanks for your comment. We have examined text and correctedEspecially, what is worthy to study that whether Ang II binding to the angiotensin type 1-receptor (AT1R) and activating membrane-bound NAD(P)H oxidase for the formation of ROS, what is worthy to study that whether or oligopeptide LSGYGP bind with AT1R through determinate the location for GABARAP. Besides, bioavailability and transport mechanisms of oligopeptide are remains to research for following bioactivity design and application of peptide drugs, these problems need to be overcome by using different design strategies in the future.” to “Future research should further discuss and confirm that how Ang II to binds the angiotensin type 1-receptor (AT1R) and activates membrane-bound NAD(P)H oxidase for the formation of ROS; how oligopeptide LSGYGP to binds with AT1R through determinate the location for GABARAP. Besides, bioavailability and transport mechanisms of oligopeptide are remains to research for following bioactivity design and application of peptide drugs, these are the key concerns attempted to overcome by using different design strategies in the future.

We have correct# indicate significantly compared with blank group (untreated cells)” to “# p < 0.001, compared with blank group (untreated cells)

We have studied reviewer’s comments carefully and have made revision which marked in red in the revised manuscript. We have tried our best to revise our manuscript according to the comments. Attached please find the revised version, which we would like to submit for your kind consideration.

We would like to express our great appreciation to you and reviewers for comments on our paper. Looking forward to hearing from you.

Thank you and best regards.

Yours sincerely

Zhong-Ji Qian, Ph.D.

Professor

Shenzhen Institute of Guangdong Ocean University, Shenzhen 518108, China

School of Chemistry and Environment, Guangdong Ocean University, Zhanjiang 524088, China

Tel: +88-759-2396270
